# Performance of FRAX in Predicting Fractures in US Postmenopausal Women with Varied Race and Genetic Profiles

**DOI:** 10.3390/jcm9010285

**Published:** 2020-01-20

**Authors:** Qing Wu, Xiangxue Xiao, Yingke Xu

**Affiliations:** 1Nevada Institute of Personalized Medicine, University of Nevada, Las Vegas, NV 89154, USA; xiangxue.xiao@unlv.edu (X.X.); yingke.xu@unlv.edu (Y.X.); 2Department of Environmental and Occupational Health, School of Public Health, University of Nevada Las Vegas, NV 89154, USA

**Keywords:** genetic risk score (GRS), bone mineral density (BMD), single nucleotide polymorphism (SNP), Fracture risk assessment tool (FRAX)

## Abstract

Background: Whether the Fracture Risk Assessment Tool (FRAX) performed differently in estimating the 10-year fracture probability in women of different genetic profiling and race remained unclear. Methods: The genomic data in the Women’s Health Initiative (WHI) study was analyzed (*n* = 23,981). The genetic risk score (GRS) was calculated from 14 fracture-associated single nucleotide polymorphisms (SNPs) for each participant. FRAX without bone mineral density (BMD) was used to estimate fracture probability. Results: FRAX significantly overestimated the risk of major osteoporotic fracture (MOF) in the WHI study. The most significant overestimation was observed in women with low GRS (predicted/observed ratio (POR): 1.61, 95% CI: 1.45–1.79) specifically Asian women (POR: 3.5, 95% CI 2.48–4.81) and in African American women (POR: 2.59, 95% CI: 2.33–2.87). Compared to the low GRS group, the 10-year probability of MOF adjusted for the FRAX score was 21% and 30% higher in the median GRS group and high GRS group, respectively. Asian, African American, and Hispanic women respectively had a 78%, 76%, and 56% lower hazard than Caucasian women after the FRAX score was adjusted. The results were similar for hip fractures. Conclusions: Our study suggested the FRAX performance varies significantly by both genetic profile and race in postmenopausal women.

## 1. Introduction

Osteoporotic fracture continues to be a critical public health problem worldwide [1,2]. One main reason is that the incidence of osteoporotic fracture increases exponentially throughout one’s life [3]. Approximately 40% of postmenopausal women will suffer at least one fracture in their lifetime [4,5,6]. Additionally, bone fractures often lead to devastating consequences, including functional decline, prolonged disability, and death [7]. With longevity increasing globally, the potentially high cumulative rate of osteoporosis and fractures, and the associated excess disability and mortality, will lead to an inevitable increase in social and economic burdens worldwide [8,9].

Furthermore, osteoporosis is a silent disease because bone loss occurs without any signs or symptoms [3]; therefore, fracture prediction becomes critically important. The Fracture Risk Assessment Tool (FRAX), which is the most widely used tool for fracture risk assessment, was developed by the Collaborating Centre for Metabolic Bone Diseases (Sheffield, UK) and is a computer-based program that computes the 10-year probability of major osteoporotic fracture (MOF, a composite of hip, humerus, forearm, and clinical vertebral fractures) and hip fracture. The FRAX can be used with or without femoral neck bone mineral density (BMD) measurement [10]. Although FRAX improves fracture prediction over the BMD T-score method alone [11], the FRAX performance of predicting fracture risk varies in different study populations [12,13,14]. Hence, there is room for further improvement in fracture prediction.

FRAX was derived from nine cohorts and has been validated in 11 independent cohorts from around the world [11]. The US FRAX was calibrated from the data of the Rochester Epidemiology Project [15] cohort, which was composed predominantly of Caucasians [16]. For the US minorities, the FRAX estimates were adjusted based on race-specific hip fracture incidence rates and race-specific mortality [17]. This adjustment was not empirically based. Racial/ethnic differences that influence fracture risk were not adequately taken into account by US FRAX [18]. Additional studies are needed to examine the performance of FRAX in US minorities.

Additionally, genetic profiling is an essential predisposition to bone deterioration and fragility fractures [19]. Genetic factors are also determinants of bone structure [20]. Although FRAX does not factor in genetic elements, mounting evidence shows that fracture susceptibility is genetically determined [21]. Virtually 50% of the variance in susceptibility to fragility fracture is attributable to genetic elements [22]. With the advancement of genomic technologies in the past two decades, major genome-wide association studies (GWASs) and genome-wide meta-analyses have successfully identified numerous genetic loci associated with fracture [15,23,24]. To date, the largest genome-wide meta-analysis on fracture, which involved 32,961 participants, revealed 14 single nucleotide polymorphisms (SNPs) associated with fracture [15]. However, the way in which these SNPs cause bone fragility and associated fracture remains unclear. As the allelic frequency of these discovered SNPs featured high variability in the population, and each SNP is associated with small effect size, the contribution of any single SNP to fracture susceptibility is expected to be minimal [25]. The cumulative effects of many associated genetic variants possibly cause osteoporotic fracture [26,27]. Thus polygenic scores summarized from risk alleles at each locus have commonly been employed to quantify the overall genetic effect contributing to fracture risk [28].

The performance of FRAX with different genetic profiling has not been reported in the literature. In addition, the performance of FRAX in minorities of the US was rarely studied. Thus our study aimed 1) to evaluate whether FRAX performs differently in estimating the 10-year absolute probability of MOF and hip fracture in postmenopausal women with different polygenic risk scores, and 2) assess FRAX performance in the prediction of MOF hip fracture in minority women. We also examined if the interaction of race and polygene score impacts the performance of FRAX in fracture prediction.

## 2. Experimental Section

### 2.1. Data Source

The Women’s Health Initiative (WHI) study is a nationwide, long-term health study that has focused on heart disease, breast and colorectal cancer, and fragility fractures in postmenopausal women. Between 1993 and 1998, the WHI has enrolled 161,808 women aged 50 to 79 years into one or more randomized clinical trials (CT) or to an observational study (OS), which were conducted at 40 clinical centers nationwide. Participants were provided by mail or telephone with questionnaires annually in the observational study, or semiannually in the clinical trials. The Institutional Review Board at each participating institution approved study protocols and consent forms [29].

### 2.2. Participants

The female participants included in the present study were combined from four WHI sub-studies: WHI Genomics and Randomized Trials Network (GARNET); National Heart Lung and Blood Institute (NHLBI); Population Architecture using Genomics and Epidemiology (PAGE); and Women’s Health Initiative Memory Study (WHIMS). These data were acquired through the database of Genotype and Phenotype (dbGap) (https://www.ncbi.nlm.nih.gov/projects/gap/cgi-bin/study.cgi?study_id=phs000200.v12.p3) with the approval of the institutional review board at the University of Nevada, Las Vegas. The included participants were genotyped using either the Illumina (Illumina Inc., San Diego, CA, USA) or Affymetrix 6.0 Array Set Platform (Affymetrix Inc., Santa Clara, CA, USA). Participants self-reported their race/ethnicity, choosing from the listed categories, which included Caucasian, African American, Hispanic, American Indian/Alaska Native, Asian, and American Indian. Participants who reported taking any medication known to influence osteoporosis, including bisphosphonates, calcitonin, parathyroid hormone, selective estrogen receptor modulators, luteinizing hormone-releasing hormone agents, and somatostatin agents, as well as participants who had incomplete information regarding risk factors included in FRAX, were excluded from the study sample. In total, 1513 subjects were excluded from the present study, and there were 23,981 eligible participants from multiple racial backgrounds, genotype data, and adjudicated fracture outcomes available.

### 2.3. Outcomes: Incident Fractures

The primary outcomes are major MOF and hip fractures. The WHI participants were followed for 9 years on average from the baseline examination. The follow-up period was calculated from the enrollment (OS) or randomization (CT) to the time of the first fracture or death. People who did not experience a fracture or death were followed until the end of the initial WHI study. Self-reported fracture outcomes were identified by questionnaires semiannually for CT participants and annually for OS participants. Radiology reports were used to adjudicate all fractures in the CT, and hip fractures in the OS. Hip fractures were centrally or locally adjudicated using the same criteria. The agreement between central and local adjudication for hip fracture was 96%. Other types of fractures were locally adjudicated at clinical centers where BMD was not measured [29].

### 2.4. Genotyping

Genotype data produced from blood samples were acquired through dbGap. Genotype imputation was completed at the Sanger Imputation Server. The Haplotype Reference Consortium (HRC) reference panel and Positional Burrows–Wheeler Transform (PBWT) imputation algorithm were used for genotype imputation. All 14 fracture-associated SNPs, as reported by Estrada et al. [15], were successfully imputed.

### 2.5. Genetic Risk Scores (GRS)

Genetic risk for fracture was quantified using a standardized metric described by Estrada et al. [15]. Briefly, this metric allows the composite assessment of genetic risk in complex traits by summarizing the genetic predisposition. Based on 14 fracture-associated SNPs discovered in the largest genome-wide meta-analysis [15], weighted genetic risk score (GRS) was calculated for each participant in this study as GRS = sum (x_i × b_i); where x_i are individual’s genotype (0, 1, 2) for SNP i, and b_i are the effect size of this SNP. Linkage disequilibrium (LD) pruning was performed in advance to eliminate possible LD that existed between SNPs. None of the 14 SNPs were removed after pruning. To illustrate the different cumulative incidence of fracture in participants with different genetic profiles, we divided the participants into three GRS groups based on their weighted GRS, using distributions of 25%, 50%, and 25%.

### 2.6. Fracture Probability

Since BMD measurement was unavailable for over 90% of the participants in this study sample, the existing FRAX score calculated by the FRAX without BMD (US FRAX version 3.0, Sheffeld, South Yorkshire, UK) for the 10-year probability of MOF and hip fracture in the data was used. The performance of FRAX with BMD according to race and GRS will be addressed in a future study. Predicted fracture probability in this study was estimated by FRAX without BMD unless otherwise specified. The observed 10-year cumulative fracture incidence was assessed by race and GRS groups. The cumulative incidence function (CIF) was applied to derive the observed 10-year fracture probability for MOF, and hip fracture accounting for competing mortality risk [30].

### 2.7. Statistical Analysis

Demographic and baseline clinical characteristics are presented as mean ± standard deviation (SD) for continuous variables or frequencies (%) for categorical variables. Differences between the two groups were examined by using Student’s *t*-test for continuous variables and by using chi-square tests for categorical variables, respectively. The ratio between FRAX predicted fracture probability and observed fracture probability (POR), with the corresponding 95% confidence interval (CI) calculated for each group. Multivariable Cox proportional hazard models were employed to assess the effect GRS and race had on survival time to the first fracture or death, with adjustment for baseline FRAX probability.

A sensitivity analysis was conducted on a small sample (*n* = 14,722) in which participants who received interventions in any of the two clinical trials, namely Calcium and Vitamin D Trial (CAD) or Hormone Therapy Trial (HT) were excluded. Statistical analyses were performed using SAS 9.4 (SAS Institute, Inc., Cary, NC, USA).

## 3. Results

### 3.1. Baseline Characteristics

The study included a total of 23,981 women for analysis. During an average of 12 years of follow-up, 5555 (23.23%) women died, and 1637 (6.9%) women sustained at least one MOF during the follow-up. Table 1 compares the baseline characteristics of women with an MOF and women without an MOF during the follow-up. Weighted GRS was significantly higher in women who sustained an MOF than in those who did not (*p* < 0.0001). Women who sustained an MOF were also older, had lower body mass index (BMI), more alcohol consumption, a higher prevalence of prior fractures, and more hip fractures in their family history. FRAX scores of both MOF and hip were significantly higher in women with a fracture incidence (*p* < 0.0001). The means of GRS between races are significantly different (Appendix A
Figure A1).

### 3.2. Performance of Fracture Risk Assessment Tool (FRAX) in Predicting Major Osteoporotic Fracture (MOF) and Hip Fracture

The crude 10-year cumulative incidence of MOF and hip fracture by the GRS group is shown in Figure 1. Significant between-group differences were observed for both MOF (*p* < 0.001) and hip fracture (*p* < 0.001). The incidences of MOF and hip fracture were greater in the high GRS group. The crude 10-year cumulative incidence of MOF and hip fracture by race is shown in Figure 2. Significant between-group differences were observed for both MOF (*p* < 0.001) and hip fracture (*p* < 0.001). The incidence of MOF and hip fracture were higher in Caucasian women.

The predicted versus observed 10-year probability of MOF by GRS groups, accounting for competing mortality, are shown in Figure 3A. The 10-year MOF probability derived from FRAX significantly overestimated risk across all GRS groups. The greatest overestimation by FRAX was observed in women who had low GRS, in which the 10-year predicted probability of MOF was 6.02% versus observed 3.74%, with a corresponding predicted/observed ratio (POR) of 1.61 (95% CI, 1.45–1.79), followed by the high GRS group with a POR of 1.38 (95% CI, 1.27–1.50), and in the median GRS group, the POR was 1.40 (95% CI, 1.32–1.49). For hip fracture outcome, the 10-year predicted probability calculated by FRAX overestimated fracture risk in all GRS groups, however, the POR was similar across the three GRS groups (Figure 3B).

The predicted versus observed 10-year probability of MOF by racial groups, with competing mortality risk accounted for are shown in Figure 4A. The 10-year probability of MOF calculated by FRAX significantly overestimated fracture risk in most racial groups, and the greatest overestimation was observed in Asian women. In Asian women, the predicted 10-year probability of MOF was 7.26% versus observed 2.03%, and the POR was 3.5 (95% CI 2.48–4.81). In African American women, the predicted 10-year probability of MOF was 3.79% as opposed to observed 1.46%, with the POR being 2.59 (95% CI 2.33–2.87). The 10-year probability of hip fracture estimated without BMD overestimated risk in all racial groups except American Indians. The 10-year predicted probability of hip fracture was in this group, with 1.75% as opposed to observed 1.91%, and with the POR being 0.91 (95% CI 0.46–1.62) (Figure 4B).

### 3.3. Race/Ethnicity and the Fracture Outcome

In the multivariate Cox proportional hazard model, after adjusting for baseline FRAX probability, weighted GRS calculated from 14 fracture-related SNPs was significantly associated with subsequent MOF. Compared to the low GRS group, the 10-year probability of MOF was 21% higher for women with medium genetic risk (hazard ratio (HR) = 1.21, 95% CI 1.05–1.39) and 30% higher for women with high genetic risk (HR = 1.30, 95% CI 1.12-1.50). Similar findings with hip fracture outcomes were observed. Compared to the low GRS group, the 10-year probability of hip fracture was 27% higher for women in the medium GRS group (HR = 1.27, 95% CI 1.04–1.55) and 46% higher for women in the high GRS group (HR = 1.46, 95% CI 1.17–1.80) (Table 2).

After controlling for baseline fracture probability estimated by FRAX, race remained a significant predictor of subsequent MOF and hip fracture. Compared to Caucasian women, Asian women had a 78% lower hazard of MOF (HR = 0.22, 95% CI 0.12–0.41) and hip fracture (HR = 0.22, 95% CI 0.09–0.52). Similarly, the FRAX-adjusted hazard ratio of MOF and hip fracture for African-American women and Hispanic women was also significantly lower (Table 3).

### 3.4. GRS and the Fracture Outcome

The potential impact of GRS on the estimated probabilities of MOF and hip fractures across different racial groups was also assessed. When adjusted for FRAX probability and race, high GRS was associated with an increased probability of MOF (GRS high vs. low: HR = 1.08, 95% CI 0.92–1.25) and of hip fracture (HR = 1.17, 95% CI 0.93–1.46) (Table 2); however, the increase was not statistically significant in both outcomes. When adjusted for the baseline FRAX probability and GRS simultaneously, the impact of race on the estimated probabilities MOF and hip fracture was slightly attenuated but remained statistically significant. Compared to Caucasian women, American Indians, Asians, African American, and Hispanic women had a 61%, 78%, 76%, and 57% lower hazard of MOF, respectively. Similar findings were observed with hip fracture outcomes (Table 3).

### 3.5. Sensitivity Analysis

We conducted a sensitivity analysis in which we excluded subjects who received intervention in either of the two clinical trials (CaD or HT). When adjusted for the FRAX probability, we observed an increased HR of MOF with GRS (GRS high as opposed to low: HR = 1.39, 95% CI 1.10–1.74); additionally, the impact of GRS on the estimated probability of hip fracture attenuated slightly (HR = 1.41, 95% CI 1.03–1.95), but remained significant. However, when adjusted for both race and FRAX probability, the association between GRS and hip fracture was not significant (Appendix A
Table A1). When controlling for GRS and FRAX probability, the effects of race on the estimated probabilities of MOF and hip fracture remained significant. Compared to Caucasian women, the race and FRAX-adjusted hazard of MOF was 90%, 78%, and 66% lower in Asian, African-American, and Hispanic women, respectively (Appendix A
Table A2).

## 4. Discussion

The present study found that FRAX overestimated the risk of fracture in women aged 50–79 years, and the degree of overestimation by FRAX in the low GRS group is greater than high genetic risk groups in both outcomes of MOF and hip fracture. In the multivariate analysis, genetic profiling was further demonstrated to be a significant predictor of MOF and hip fracture, independent of FRAX probability.

Genetic factors that influence osteoporotic fracture risk have long been recognized. Genetics are determinants of bone structure and thus a predisposition to fragility. Hereditary factors contribute almost half of the variance in fracture susceptibility [22]. However, genetic factors are not counted in the FRAX or any other existing clinical fracture risk assessment model. Since FRAX is the most commonly used fracture prediction model, determining if the performance of FRAX varies with different genetic profiling has become crucially important. The largest and most updated GWAS meta-analysis has identified 14 SNPs that are significantly associated with fracture risk at a significant genome-wide level [22]. Although these individual SNPs have modest effect size on fracture risk, the GRS, as summarized from these individual risk SNPs, enables us to examine if FRAX performance varies with different genetic risk factors. The varied prediction performance of FRAX by GRS, as observed in our study, suggests that the accuracy of FRAX can be improved by incorporating genetic profiling. Several studies suggested that including GRS as a predictor may help improve the accuracy of various fracture prediction models. For example, GRS of 39 SNPs increased the precision of non-vertebral fracture prediction in postmenopausal Korean women [31]. Additionally, GRS based on 63 SNPs improved the accuracy of non-trauma fracture prediction [26]. One of our recent studies on older US men also found that GRS is one of the most important variables in MOF prediction models developed by the gradient boosting approach [32].

The present study also provides compelling evidence that FRAX overestimates the risk of MOF and hip fracture in women 50–79 years old, across all racial groups, but especially in minorities. In Asian, African-American, and Hispanic women, the observed probability of fracture, in terms of both MOF and hip fracture, was significantly lower than the risk estimated by FRAX, indicating that the FRAX did not adequately capture racial and ethnic differences of fracture risk. Additionally, our multivariate analysis demonstrated that race is a significant predictor of MOF and hip fracture independent of the cumulative fracture risk estimated by FRAX, suggesting that FRAX does not have adequate adjustment for racial difference. Racial and ethnic difference that influences fracture risk not being adjusted for adequately in the FRAX has long been a concern [10]. As we know, the US FRAX was calibrated from the Rochester Epidemiolofy Project data, composed predominantly of Caucasians. For non-Caucasian minorities, the FRAX estimates were adjusted based on race-specific hip fracture incidence rates and race-specific mortality [33]. This adjustment for minorities in FRAX is not empirically based, thus making the prediction accuracy of FRAX increasingly uncertain, especially for MOF, a composite of hip, humerus, forearm, and clinical vertebral fractures. The current FRAX adjustment model, based on race-specific hip fracture incidence rate and race-specific mortality, remains likely to be inadequate for MOF risk estimation in minorities. In this study, we observed that the overestimated risk for MOF by FRAX was much higher than that for hip fracture, which validated that the US FRAX has not adjusted race adequately for MOF. A prior study conducted on the same WHI sample assessing the accuracy of FRAX without BMD in predicting fracture also demonstrated that the FRAX has significantly lower sensitivity in identifying incidence fractures in African-American and Hispanic women [34]. Another study on 2266 postmenopausal women who participated in the Hong Kong osteoporosis study revealed that the predictive accuracy of FRAX with BMD was not substantially different from the model with BMD alone [35]. Considering the generally lower incidence of fracture in Asians than in Caucasians, as well as the prominent effect of BMD in fracture prediction in the Asian population, the absence of BMD in the present study may explain the significant overestimation of fractures in this racial group. Moreover, inconsistent findings regarding the performance of FRAX without BMD was reported in several other studies. Leslie et al. observed that the fracture probability estimated without BMD overestimate risk among the general population [36], which is consistent with findings from the present study. Other studies have reported underestimation of fracture risk by FRAX [12,37,38], but their methodologies have lately been found to be problematic because they either compared incidence with probabilities or failed to take the competing mortality risk into account [39].

When both FRAX probability and race were adjusted simultaneously in the multivariate model, the effect of GRS was reduced, which could be due to the following reasons. First, genetic profiling regarding osteoporosis or osteoporotic fracture varies in different racial groups; the effect of race and GRS on fracture could be overlapping (See Appendix A
Figure A1). Second, the genetic effect on fracture probability may not be fully captured by the limited number of discovered risk SNPs. With more fracture-related genetic components being discovered in the future, a larger effect of GRS on fracture risk prediction can be foreseen.

Limitations to this study are acknowledged. First, the WHI data we used only included women aged 50–79 years, so our findings may not apply to men or to women who are not in the study age range. Second, rare genetic variants with high effect size were not included in the present study, mainly because risk SNPs used in this analysis were identified from a GWAS meta-analysis study, which likely discovered common but not rare variants [15]. The limited number of fracture-associated SNPs may not capture all genetic risk, which partially explained the reduced effect of GRS in the model when both FRAX probability and race were included. Third, our study only focuses on FRAX without BMD because the BMD measurement was unavailable for most of the study subjects. The performance of FRAX with BMD will be examined in a future study. Finally, the sample size of Asian and American Indian subjects was very small in this study; the results may, therefore, be underpowered.

## 5. Conclusions

To the best of our knowledge, this is the first study to assess FRAX performance in the prediction of MOF and hip fractures in groups with different genetic profiling and of various races. Our findings suggested genetic profiling of an individual should be considered in fracture prediction, as genetic factors have been demonstrated to be a significant risk factor for osteoporotic fracture, independent of FRAX. Our results also demonstrated that FRAX performed differently in different races, and thus the effect of race in osteoporotic fracture prediction has not heretofore adequately been taken into account by existing FRAX models. Fully integrating genetic profiling and racial factors into the existing fracture assessment model is very likely to improve the accuracy of fracture prediction. Thus, developing racial/ethnic-specific, individualized fracture risk-assessment models will provide more accurate fracture risk assessment. Further studies, especially those including men, larger samples of minorities, and more comprehensive fracture-associated genetic variants, are clearly warranted.

## Figures and Tables

**Figure 1 jcm-09-00285-f001:**
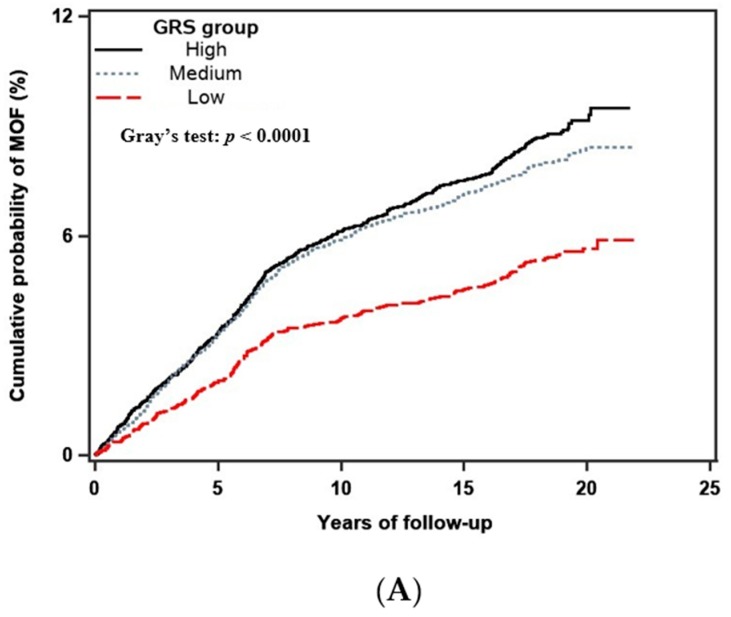
Crude (unadjusted) 10-year cumulative incidence of major osteoporotic (**A**) and hip fracture (**B**) stratified by the GRS group, including competing mortality risk. The difference in the cumulative incidence rates among different GRS groups was tested by using Gray’s test, with *p*-value <0.01 indicating a significant difference between the groups.

**Figure 2 jcm-09-00285-f002:**
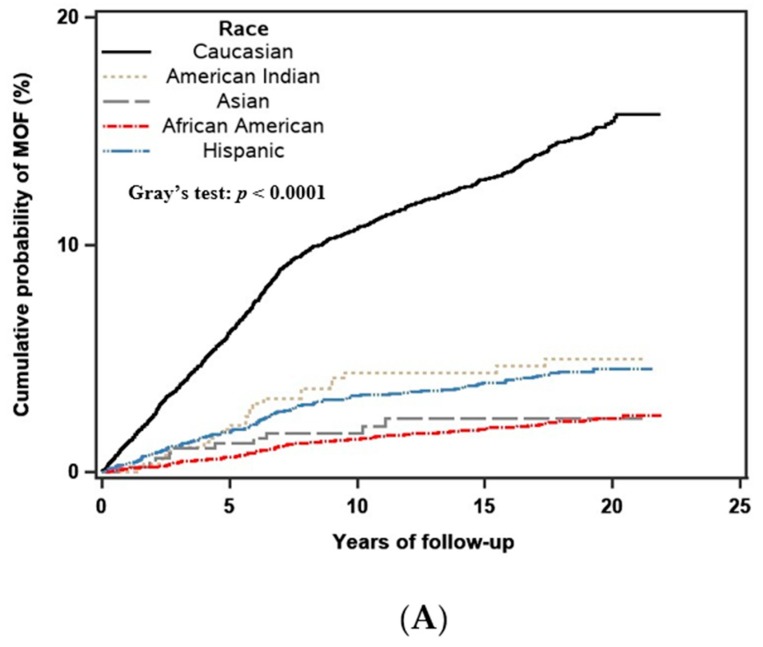
Crude (unadjusted) 10-year cumulative incidence of major osteoporotic (**A**) and hip fracture (**B**) stratified by race, including competing mortality risk. The difference in the cumulative incidence rates among different racial groups was tested by using Gray’s test, with *p*-value <0.01 indicating a significant difference between the groups.

**Figure 3 jcm-09-00285-f003:**
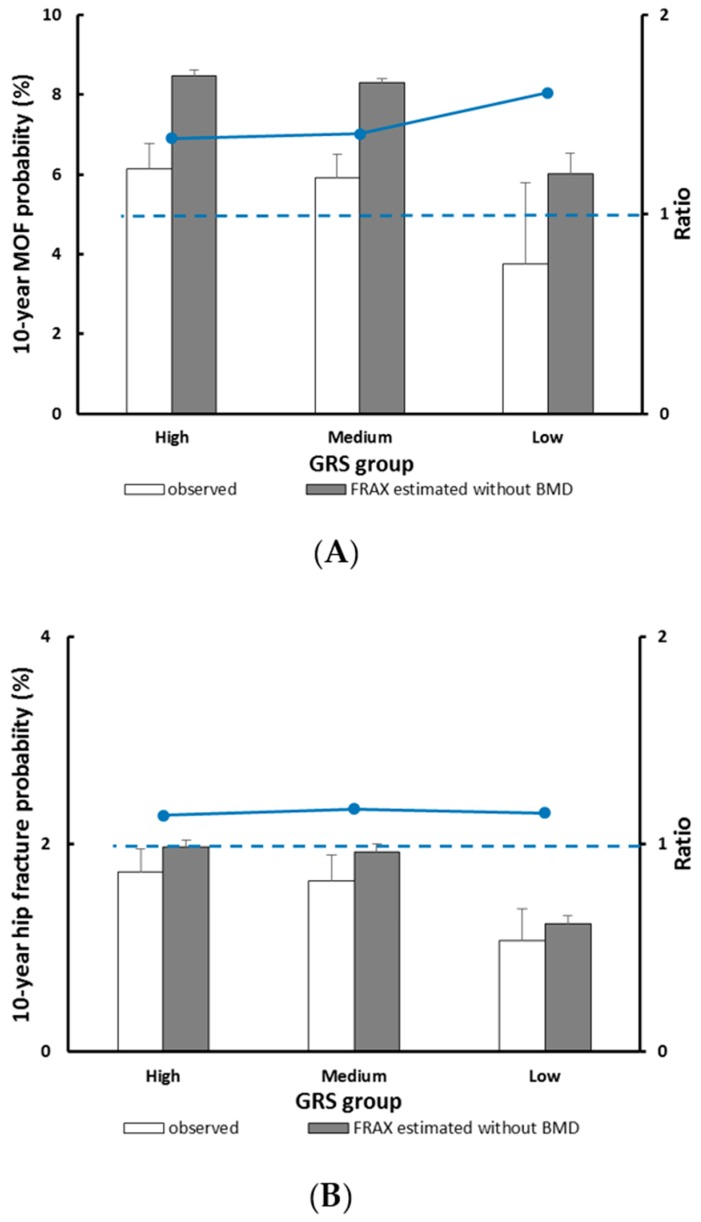
Observed versus predicted 10-year major osteoporotic fracture (**A**) and hip fracture (**B**) probability stratified by the GRS group. The dotted line indicates a relative ratio of 1 (reference line); ratio >1 indicates that FRAX overestimates fracture probability.

**Figure 4 jcm-09-00285-f004:**
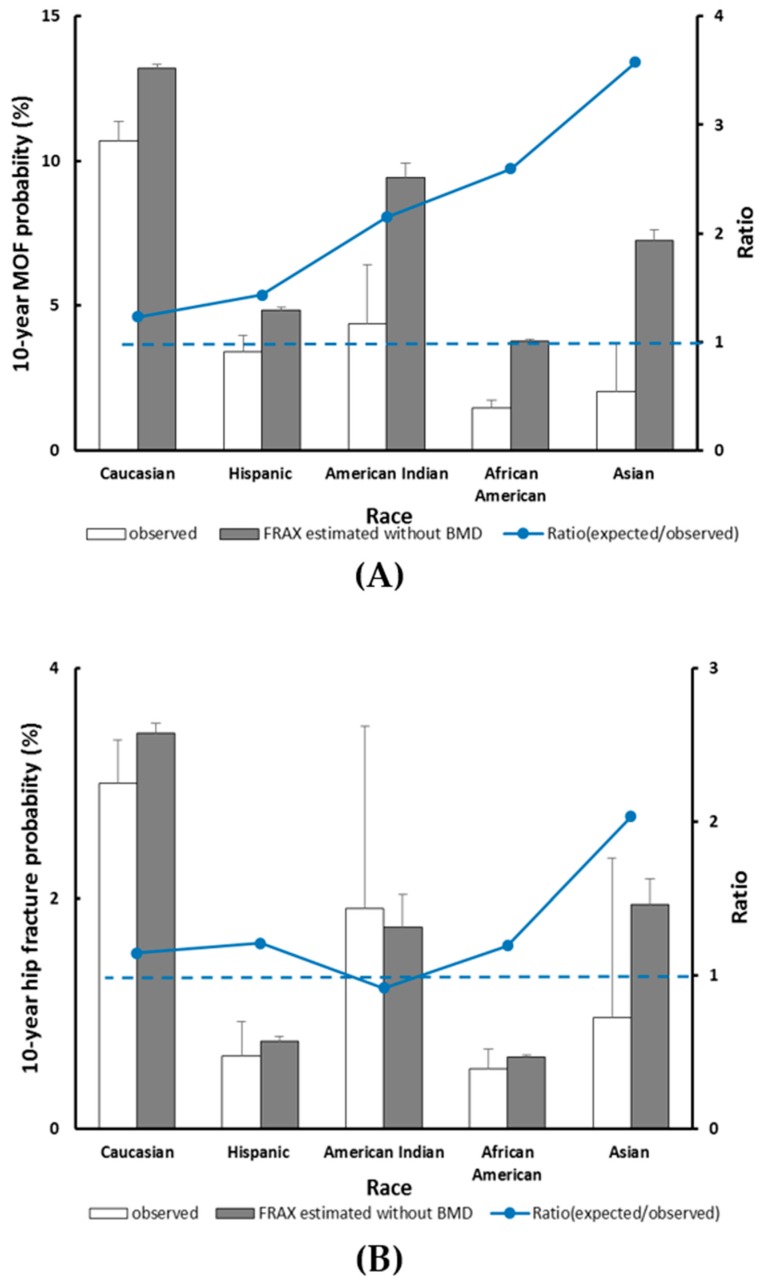
Observed versus predicted 10-year major osteoporotic fracture (**A**) and hip fracture (**B**) probability stratified by race. The dotted line indicates a relative ratio of 1 (reference line); ratio >1 indicates that FRAX overestimates fracture probability.

**Table 1 jcm-09-00285-t001:** Baseline characteristics of 23,981 women according to whether they sustained a major osteoporotic fracture (MOF) during follow-up.

	Subjects with Major Osteoporotic Fracture Event (*n* = 1637)	Subjects without Major Osteoporotic Fracture Event (*n* = 22,281)	*p*-Value
**Age (year), Mean ± standard deviation (SD)**	67.99 ± 6.52	63.26 (±7.32)	**<0.0001**
**Weight (kg), Mean ± SD**	73.59 ± 15.21	77.32 (±16.92)	**<0.0001**
**Height (cm), Mean ± SD**	161.25 ± 6.30	161.06 (±6.29)	0.28
**Body mass index (kg/m^2^), Mean ± SD**	28.27 ± 6.30	29.73 (±6.09)	**<0.0001**
**Smoking, *n* (%)**			0.35
Never	858 (52.42)	11,704 (52.52)	
Past	639 (39.03)	8448 (37.92)	
Current	140 (8.55)	2129 (9.56)	
**≥3 alcoholic drinks per day, *n* (%)**			0.05
Yes	24 (1.47)	216 (0.97)	
No	1613 (98.53)	22,065 (99.03)	
**Rheumatoid arthritis, *n* (%)**			0.91
Yes	109 (6.66)	1500 (6.73)	
No	1528 (93.34)	20,781 (93.27)	
**Previous fragility fractures, *n* (%)**			**<0.0001**
Yes	835 (51.01)	6902 (30.98)	
No	802 (48.99)	15,379 (95.04)	
**Familial history of hip fracture, *n* (%)**			**<0.0001**
Yes	271 (16.55)	2156 (9.68)	
No	1366 (83.45)	20,125 (93.64)	
**Race, *n* (%)**			
**Caucasian**	1255 (76.66)	7948 (35.67)	**<0.0001**
**American Indian**	24 (1.47)	535 (2.40)	
**Asian**	10 (0.61)	467 (2.10)	
**African American**	189 (11.55)	9231 (41.43)	
**Hispanic**	159 (9.71)	4100 (18.40)	
**Genetic risk score (GRS), Mean ± SD**	0.58 ± 0.12	0.56 ± 0.13	**<0.0001**
**Fracture Risk Assessment Tool (FRAX^®^) for MOF (%), Mean ± SD**	13.51 ± 8.57	7.39 ± 6.27	**<0.0001**
**FRAX^®^ for hip fracture (%), Mean ± SD**	4.02 ± 5.45	1.61 ± 2.88	**<0.0001**

GRS: genetic risk score calculated based on 14 fracture-related single nucleotide polymorphism (SNPs). Significant results are in boldface.

**Table 2 jcm-09-00285-t002:** Hazard ratios (HR) with 95% confidence interval (CI) for outcomes of incidence fracture according to the GRS group, adjusted for FRAX score: Results of multivariate Cox proportional hazard model.

	Major Osteoporotic Fracture	Hip Fracture
	HR (95% CI)	HR (95% CI)
**Adjusted for FRAX probability**		
**low**	1(reference)	1(reference)
**medium**	**1.21 (1.05–** **1.39)**	**1.27 (1.04–** **1.55)**
**high**	**1.30 (1.12–** **1.50)**	**1.46 (1.17–** **1.80)**
**Adjusted for FRAX probability + race**		
**low**	1(reference)	1(reference)
**medium**	1.01 (0.88–1.16)	1.00 (0.81–1.22)
**high**	1.08 (0.92–1.25)	1.17 (0.93–1.46)

Significant results are in boldface.

**Table 3 jcm-09-00285-t003:** Hazard ratios (HR) with 95% confidence interval (CI) for outcomes of incidence fracture according to race, adjusted for FRAX score: Results of multivariate Cox proportional hazard model.

	Major Osteoporotic Fracture	Hip Fracture
	HR (95% CI)	HR (95% CI)
**Adjusted for FRAX probability**		
**Caucasian**	1 (reference)	1 (reference)
**American Indian**	**0.40 (0.26–0.59)**	**0.39 (0.21–0.70)**
**Asian**	**0.22 (0.12–0.41)**	**0.22 (0.09–0.52)**
**AA**	**0.24 (0.20–0.28)**	**0.22 (0.17–0.27)**
**Hispanic**	**0.44 (0.37–0.52)**	**0.25 (0.20–0.34)**
**Adjusted for FRAX probability + GRS group**		
**Caucasian**	1 (reference)	1 (reference)
**American Indian**	**0.39 (0.26–0.59)**	**0.38 (0.21–0.68)**
**Asian**	**0.22 (0.12–0.40)**	**0.20 (0.09–0.49)**
**AA**	**0.24 (0.20–0.29)**	**0.20 (0.18–0.28)**
**Hispanic**	**0.43 (0.36–0.52)**	**0.24 (0.18–0.32)**

Significant results are in boldface.

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
