# Peer review of "Performance of FRAX in Predicting Fractures in US Postmenopausal Women with Varied Race and Genetic Profiles"

_jcm, 2020, doi:10.3390/jcm9010285_

Round 1

Reviewer 1 Report

This study was undertaken to ascertain if the addition of race and/or genetic risk score can improve the fracture prediction accuracy of FRAX. It is well established that FRAX suffers from limitations and identification of additional factors that can be augment FRAX to better predict fracture risk is a worthy endeavor. Strengths include a large sample size with adjudicated fractures and the SNP analysis. Weaknesses are well described; in particular the lack of BMD data is unfortunate but understandable given the sample. However several issues need to be addressed before this manuscript can be recommended for publication.

Major Concerns        

The SNP analysis is predicated on calculating the genetic risk score developed by Estrada et al. It would be interesting to instead independently build a model to weight the influence of these SNPs on the fractures reported in this study. This is especially important because the Estrada study only evaluated European and East Asian subjects and the authors of this study are trying to make a point of the importance of considering race. How was race classified? Self report? What about subjects from multiple race groups? In addition, the number for each category is not presented. With the authors indicating that there were small sample sizes for the Asian and American Indian subjects, these numbers are important. Table 1 compares the baseline characteristics between subjects with and without MOF. This analysis should be expanded to also compare between the race categories using a multivariate model. It is premature to conclude that the observed differences in FRAX, GRS, and MOF are based on race without assessing these potential covariates.

Minor Concerns

Citing a manuscript under review in inappropriate. The entire manuscript needs to be edited for proper English.

Author Response

This study was undertaken to ascertain if the addition of race and/or genetic risk score can improve the fracture prediction accuracy of FRAX. It is well established that FRAX suffers from limitations and identification of additional factors that can be augment FRAX to better predict fracture risk is a worthy endeavor. Strengths include a large sample size with adjudicated fractures and the SNP analysis. Weaknesses are well described; in particular the lack of BMD data is unfortunate but understandable given the sample. However several issues need to be addressed before this manuscript can be recommended for publication.

Authors’ response: Thank you for the comments.

The SNP analysis is predicated on calculating the genetic risk score developed by Estrada et al. It would be interesting to instead independently build a model to weight the influence of these SNPs on the fractures reported in this study. This is especially important because the Estrada study only evaluated European and East Asian subjects and the authors of this study are trying to make a point of the importance of considering race.

Authors’ response: Thanks for the suggestion, and we agree with the reviewer. Genetic risk score is commonly used to integrate and quantify the effect of SNPs (S. A Lambert et al., Towards clinical utility of polygenic risk scores, Human Molecular Genetics, 2019). The score is calculated by summing risk alleles and weighted by effect sizes derived from GWAS results (Wray, N. R., et al., Prediction of individual genetic risk to disease from genome-wide association studies, Genome Res., 2007). In addition, GRS derived from these SNPs was demonstrated to be associated with lower BMD in African Americans (X. Xiao et al., Association between a literature-based genetic risk score and bone mineral density of African American women in Women Health Initiative Study, Osteoporos Int. 2019). In this study, we are focusing on the different performance of FRAX in different GRS and race groups. The influence of these SNPs on fracture will be examined with further study in the future.

How was race classified? Self report? What about subjects from multiple race groups?

Authors’ response: Thank you for the question. At baseline, WHI participants self-reported their race/ethnicity, choosing one from the listed categories, which included Caucasian, African American, Hispanic, American Indian/Alaska Native, Asian, and American Indian. We added the corresponding content to the experimental section. (Please see 2.2 participants)

In addition, the number for each category is not presented. With the authors indicating that there were small sample sizes for the Asian and American Indian subjects, these numbers are important.

Authors’ response: Thank you for the suggestion. In the revision, we followed the reviewer’s recommendations and examined the frequency of each race category.  In this study there were 9,203 Caucasian (with MOF: 1,255, without MOF: 7,948), 559 American Indian (with MOF: 24, without MOF: 535), 477 Asian (with MOF: 10, without MOF: 467), 9,420 African American (with MOF: 189, without MOF: 9,231), and 4,259 Hispanic (with MOF: 159, without MOF: 4,100). (Please see Table 1).

Table 1 compares the baseline characteristics between subjects with and without MOF. This analysis should be expanded to also compare between the race categories using a multivariate model. It is premature to conclude that the observed differences in FRAX, GRS, and MOF are based on race without assessing these potential covariates.

Authors’ response: Thanks for your suggestions. In this study, we are evaluating if FRAX can fully capture the fracture risk for people in different GRS and race groups. In the multivariate analysis with MOF as an outcome, we adjusted for the FRAX score to see if the variable of GRS or race groups is significant. If yes, we conclude the FRAX perform differently among the subgroup.  Please note that important clinical risk factors for fracture, including age, sex, weight, height, previous fracture, parental hip fracture, current smoking, glucocorticoids, rheumatoid arthritis, secondary osteoporosis and alcohol intake (≥ 3units/day), are included in the FRAX algorithm.

Citing a manuscript under review in inappropriate. The entire manuscript needs to be edited for proper English. 

Authors’ response: W agree with the reviewer.  The citation of this work will be available in the next few days. We will contact the Journal publisher to find an appropriate way to cite this work. Also, we have revised the whole manuscript carefully and corrected all grammar or syntax errors.

Reviewer 2 Report

Major comments

Authors performed a retrospective analysis of data from the WHI study evaluating performance of FRAX in predicting fracture risk accross a population with different etnicity and genetic profile. Performance of  individuals' genetic profile in fracture risk prediction was assessed, as well. Data are interesting and I have suggestion for improving reading of the paper and conclusions. 

Introduction

-This section is too long. I suggest to significantly shorten it. For example, the second paragraph is not necessary and could be summarized in a couple of sentences.  

Experimental section

-It is not clear whether patients on anti-osteoporosis medications or on any drug possibly influencing bone metabolism (eg glucocorticoids) were excluded.

Risulta

-Please cancel the first sentenceof this section

-I suggest to show number and percentages of women for any etnicity in Table 1.

Discussion

-Page 9, lines 282-284. Here and in the conclusion section authors suggest that 'incorporating genetic profiling' would improve accuracy of fracture prediction. It would be of utmost interest for the reader to give an idea of how it could be done in clinical practice.  Familiar history of hip fracture is just included in the FRAX and has high influence in the algorithm. Would the authors suggest to include the presence of any MOF and/or postmenopausal osteoporosis among parents and/or other relatives? Would propose to perform further studies in this sense?

Author Response

Authors performed a retrospective analysis of data from the WHI study evaluating performance of FRAX in predicting fracture risk across a population with different ethnicity and genetic profile. Performance of individuals' genetic profile in fracture risk prediction was assessed, as well. Data are interesting and I have suggestion for improving reading of the paper and conclusions. 

Authors’ response: Thanks for your comments.

Introduction

This section is too long. I suggest to significantly shorten it. For example, the second paragraph is not necessary and could be summarized in a couple of sentences.  

Authors’ response: Thanks for your advice, and we have revised the second paragraph (Please see Introduction).

Experimental section

It is not clear whether patients on anti-osteoporosis medications or on any drug possibly influencing bone metabolism (eg glucocorticoids) were excluded.

Authors’ response: Thanks for your question. In our analysis, we excluded participants who reported taking any medication known to influence osteoporosis, including bisphosphonates, calcitonin, parathyroid hormone, selective estrogen receptor modulators, luteinizing hormone-releasing hormone agents, and somatostatin agents, and participants who had incomplete information regarding risk factors included in FRAX. (Please see 2.2 Participants)

Results

Please cancel the first sentence of this section. I suggest to show number and percentages of women for any ethnicity in Table 1.

Authors’ response: Thanks for the suggestion. In the revision, we deleted the first sentence of this section.  We followed the reviewer’s recommendations and examined the frequency of each race category.  In this study, there were 9,203 Caucasian (with MOF: 1,255, without MOF: 7,948), 559 American Indian (with MOF: 24, without MOF: 535), 477 Asian (with MOF: 10, without MOF: 467), 9,420 African American (with MOF: 189, without MOF: 9,231), 4,259 Hispanic (with MOF: 159, without MOF: 4,100). (Please see Table 1).

Discussion

Page 9, lines 282-284. Here and in the conclusion section authors suggest that 'incorporating genetic profiling' would improve accuracy of fracture prediction. It would be of utmost interest for the reader to give an idea of how it could be done in clinical practice. 

Authors’ response: Thanks for the comments. Several studies suggested that including GRS as a predictor may help improve the accuracy of various fracture prediction models. We revised the corresponding content in our discussion (Please see Discussion).

Familiar history of hip fracture is just included in the FRAX and has high influence in the algorithm. Would the authors suggest to include the presence of any MOF and/or postmenopausal osteoporosis among parents and/or other relatives? Would propose to perform further studies in this sense?

Authors’ response: Thanks for your comments. We agree with the reviewer that a family history of hip fracture is important, which supports our current study, genetic components are important for fracture prediction Whether to include the presence of any MOF and/or postmenopausal osteoporosis among parents and/or other relatives into the algorithm needs additional research, and it would be an interesting topic in the future study.